# Telework and Social Services in Spain during the COVID-19 Pandemic

**DOI:** 10.3390/ijerph18020725

**Published:** 2021-01-15

**Authors:** Aleix Morilla-Luchena, Rocío Muñoz-Moreno, Alfonso Chaves-Montero, Octavio Vázquez-Aguado

**Affiliations:** Department of Sociology, Social Work and Public Health, Faculty of Social Work; COIDESO Research Centre, Contemporary Thought and Innovation for Social Development; University of Huelva, 21007 Huelva, Spain; aleix.morilla@dstso.uhu.es (A.M.-L.); rocio.munoz@dstso.uhu.es (R.M.-M.); octavio@dstso.uhu.es (O.V.-A.)

**Keywords:** teleworking, social services, COVID-19, quality of working life, job satisfaction

## Abstract

This paper analyses teleworking in social services during the state of alarm caused by the COVID-19 pandemic in Spain. It has a double objective: To analyse the profile of the professional who teleworked in social services and, on the other hand, to analyse the perception of teleworkers of working conditions during this period, as well as the degree to which they have been affected by them depending on whether they work face-to-face or telematically. To this end, a questionnaire was administered to Spanish social service professionals working, obtaining a sample of 560 professionals in the sector. The profile obtained in relation to teleworking may be especially useful when considering the progressive incorporation of more non-presential activity in social services, and the results show that, although teleworking has been perceived as an efficient way of overcoming the limitations to face-to-face work arising from the pandemic, both the positive and negative consequences of the implementation of this modality of work should be carefully assessed.

## 1. Introduction

The impact of the COVID-19 pandemic has been remarkable in all aspects of life. From the economy to social interaction, its impacts have been significant. Social services, as a system of social protection for large segments of the population, have not been exempt to this impact they have already pointed out [1,2]. This article aims to explain how social service professionals have adapted to this reality. Moreover, it is important to understand the assessment they make of the teleworking experience they have developed during the existence of the state of alarm in Spain.

As is well known, on 14 March 2020 a state of alarm was declared in Spain, as a means of dealing with the health repercussions of the pandemic caused by the first wave of the COVID-19. If this strategy served to curb the impact of the pandemic on the Spanish health system at first, it is no less true that it had serious consequences for all economic and social life in our country. Suddenly, with the exception of the activities considered essential, all the others were limited in their possibilities of implementation. The Spanish social services were not unaware of this impact of the state of alarm. Suddenly, all the activities of social intervention, assistance and support to families were limited and interrupted with serious repercussions for the target population. Such was the impact that only 12 days later the Spanish authorities, by order SND/295/2020, declared the social services and their workers to be essential, which made it possible to continue providing these services, albeit conditioned by the state of alarm. In this way, social services professionals have had to adapt their professional practice to the new circumstances, limiting contacts and interactions, increasing health and safety measures and incorporating teleworking into their activity in order to comply with the protection measures in place, and relegating direct, face-to-face attention, which could be identified as one of their signs of identity, to cases where it was strictly necessary.

This new situation poses a challenge for social services. This system is based on a high relational component, which means that the quality of the service is basically in the hands of the people who interact directly with the recipients of the service already pointed out [3]. The search for quality, modernization and improvement of Social Services has been considered as an efficient strategy, contrary to the classic factor of “more resources”, more expenditure, more staff, more means of all kinds they have already mentioned [4].

Although these two strategies are compatible (and in fact, it would be desirable for them to occur simultaneously), perhaps in the present and most immediate future marked by the COVID-19 pandemic. It seems that a new crisis is about to be faced, and the true magnitude of which is still difficult to quantify.

All this will result in practice in Social Services, which adjusted to a bureaucratic formality and constantly overwhelmed by demand, can be almost nothing more than managers of economic benefits (direct or indirect) at the service of individuals and families—not groups or collectivities—who are in a position to demand them already pointed out [5]. The Third Report on Social Services in Spain [6] states that the main problems of these professionals, include the excessive workload, stress and saturation (39.9%), the high level of bureaucracy (16.2%) and the existence of few resources (14.5%).

This same report identifies, as the main area of social services in which they carry out their professional activity, that of “information, guidance, counselling, diagnosis and assessment” (49.2%, practically half of the professionals), followed at a great distance by the areas of “personal autonomy, home care and family respite” (9.4%), “intervention and family support” (8.4%) and the area of “management” (8.3%). In this way, the prevalence of care and bureaucratic functions is noted, above others such as direct group care, prevention, promotion and social insertion, and with little involvement in research they have already signaled [7].

In addition, more than one in four professionals (26.3%) consider that the physical conditions of their work space are “bad” or “very bad”: Shared offices, problems associated with lighting and/or temperature, architectural barriers, etc., thus, constituting uncomfortable spaces for the people who attend and for themselves.

### What Role Can Telework Play in Social Services?

In this context, and given the characteristics of the professional practice of social services, it is increasingly necessary to work through technological tools that allow mass mailings to citizens about campaigns, courses, etc., that may be of interest to them. Similarly, secure means of consultation must be established, both for files and general information, which allow agile and transparent information on benefits and services channelled mainly by telematic means, and if not possible, by telephone, to the detriment of face-to-face, which is extremely inefficient and slow if the objective is merely informative already pointed out [8].

In the words of [9], “teleworking consists of the habitual performance of remote work activity in a different place from the usual place in the company, and sometimes at different times” (p. 2). It can also be carried out from home to try to reconcile family life or in coworking. Teleworking requires Internet access and technological equipment in order to carry out work tasks satisfactorily. This new working method provides greater time flexibility and can offer advantages for companies and public institutions, as well as for workers, although it is easier to implement in some jobs than in others.

In the context of the COVID-19 pandemic, the situation may lead to an increase in teleworking, not only as an occasional measure resulting from environmental or health disasters, but also as a strategy to reduce infrastructure costs, in order to reduce pollution related to mobility or to generate a favourable climate for combining work and family life they have already highlighted [10].

In Spain, only 4.8% of employees were teleworking in 2019 although, according to some surveys, this percentage would have reached 34% of people teleworking during the weeks of confinement due to the COVID-19 crisis. Although the conditions in which this massive practice has occurred have not been the most appropriate, they have made it possible to highlight its viability and allow it to promote the implementation of this type of work in those companies and cases in which it is productive, effective and satisfactory for workers and managers [9]. In fact, the national government issued a Royal Decree-Lawpoints to [11] in September 2020 aimed at regulating this labour practice in Spanish companies.

Prior to the COVID-19 crisis, if the number of people teleworking was already generally low, this percentage is equally or even lower in the Social Services sector where, as mentioned, due to its high relational component and the way it has been functioning, it is unlikely that an attempt has been made to implement this modality in a generalised way in the organisation beyond the fact that it has been used in some specific cases. And yet, this exceptional circumstance may have “forced” many of these workers to adapt to the new context and incorporate teleworking into their professional practice, showing that perhaps, unlike what might be thought, and although not in the totality of the work, it is possible to use this means for many of the tasks that they have been carrying out.

It can be said that teleworking is a rapidly growing work practice, but its effects on the psychological well-being of employees have not been well-studied. This psychological well-being refers, in a general way, to the positive feeling and constructive thinking of the human being about himself, which is defined by his subjective experiential nature and which is closely related to particular aspects of physical, psychic and social functioning. In this way, a greater psychological well-being can lead to be more productive, have better responses to cope with stress, greater sociability, among other elements they have already identified [12].

Some of these elements, among many others, that could influence the psychological well-being of individuals have been studied in relation to teleworking, such as organiza-tional performance, which following [13], would be understood as the organization’s abil-ity to achieve its objectives by using resources efficiently and effectively. Moreover, it is also considered job satisfaction or dissatisfaction already pointed out [14] as reactions, sensations and feelings of a member of the organization towards its work, commitment already mentioned [15], such as the degree to which an employee identifies with a particular organization and its objectives, and wishes to maintain its relationship with it or social support, understood as the feeling of being appreciated and valued by others and of belonging to a social network already pointed out [16]. Although it is not the object of this study to delve into these elements, it is considered of interest to mention them and discuss some of them, in a general way, in order to provide a richer vision in addition to the descriptive elements obtained through the ad hoc questionnaire used in this work.

Along these lines, authors, such as [17] already highlighted that teleworking can mean the possibility of greater independence, but the profound change it implies in working relations and conditions that can mean isolation and a greater burden for those who telework cannot be ignored. It could also be said that it has the positive effect of opening up great possibilities for sectors that have usually had barriers to access to work, such as people with disabilities or those who have people in their care (often women who care for the dependent population). In these cases, there is a benefit in the possibility of accessing the labour market, while continuing with their family activity. On the other hand, teleworking also represents an opportunity for men to start taking on more actively their parenting or caring for dependent older people, without moving away from their usual work activities already indicated [18].

In this way, teleworking is presented as a particularly interesting element in favouring harmony between work and private life already specified [19], and has been studied in relation to its possibilities for expanding the labour market, job flexibility, inclusion of the population in a situation of disability in working life and reconciliation of family life they have already pointed out [20]. Telework is recognised as an element favouring progress towards an effective balance between work, family and personal life already pointed out in [21]. It is particularly interesting, considering a traditionally feminized sector, such as social services, where the presence of women is still much higher they have already indicated [6,22,23], but without forgetting the possible psychosocial risk factors which may arise. These range from difficulties in combining work and family to avoiding overlapping of work and household tasks (greater likelihood of interruptions during working time). The need to delimit the physical space of work when it is done at home; the lack of distinction between time and space of work; family and social life; complications in managing schedules and food or the neglect of the person himself already noted [24]. It is therefore clear that teleworking has enormous potential to have a positive impact on worker satisfaction, but also on worker dissatisfaction.

In addition to the aforementioned isolation, and especially in the context of a feminised sector, although there could be an increase in personal fulfilment as both home and work needs can be met, there is a fear of the impact that teleworking may have on career development within the organisation. Therefore, the trust of superiors in this medium is necessary as a legitimate form of work organization and taking into account teleworkers for promotions and employment advancement in the organization under the same conditions as full-time workers they have already pointed out [25]. On the other hand, the difficulties in applying the Law on Prevention of Occupational Risks and the duty of organisations to guarantee the safety of their workers clashes with elements, such as the right to the inviolability of the home, the mobile reconfiguration of the workplace or the technical-preventive regulations on the skills needed to carry out risk assessment they have already signaled [26].

If communication technologies and legal advances are making telework more widespread for certain productive sectors and for a specific typology of workers, this article explores the impact that telework has had on social services in times of confinement, as well as we are interested in the assessment that professionals affected by its development make of it. Let us remember that this is a professional activity where, on the one hand, human contact and interaction has been privileged and, on the other hand, not all the population to which its activities and services are addressed have the means to make telematic contact with the system possible. It should be added that the generalisation of teleworking in the social services would be a real transformation within the system to which both professionals and the population in general could object. In this context, the aim of this research has been to analyse how professionals have perceived the experience of teleworking in the field of social services and what characteristics define the professionals who have carried out their work through this modality in Spain during the first wave of the COVID-19.

## 2. Materials and Methods

The results shown in this paper are based on a questionnaire administered to social service professionals, based throughout Spain. A total of 560 professionals from the sector have participated. The fieldwork was carried out between 1 and 19 April 2020, i.e., in the midst of the confinement and alarm situation in Spain.

Given the health emergency produced by the pandemic and the situation of confinement during the period in which the fieldwork is carried out, an online tool was created through google forms, which would facilitate the collection of information from social services professionals. The global pandemic situation has limited the traditional condi-tions of field work in social research and has forced the academic world to rethink and in-troduce new methodologies that allow access with greater guarantees to the population under study.

In addition to the inability to access respondents during the alarm state, the admin-istration of this online questionnaire has other important qualities, such as cost reduction, the ease of access to it through any electronic device, as well as the possibility of reaching a larger population under study in a shorter period of time (See Appendix A).

Different authors [27,28] endorse the use of this information collection tool, since it has proven to be methodologically practical, suitable and especially useful for obtaining a satisfactory statistical product. In addition, it has been applied in other studies.

As for the way of access to the population under study, a multiple invitation protocol is applied through a snowball process. The questionnaire has in turn been disseminated through social networks and shared from different professional associations, thus ensuring that this instrument reaches the greatest possible number of professionals within this field. Although a stratified sampling by quotas has not been established, given the difficulty of obtaining a detailed census of social service professionals in Spain, and also in order to reach the greatest possible number of responses, all the autonomous communities are represented.

Although it has been sought to guarantee the maximum representativeness of the Spanish territory, the results of this work are not intended to be fully generalized to the whole population, but also make an approach to this reality through the description and analysis of the sample.

This work complies with the requirements of the Ethics Committee of the Vice Rector’s Office for Research of the University of Huelva. As it is not considered experimental research, no specific authorisation document is required.

The questionnaire administered consists of five different parts: A first part on the socio-demographic characteristics and professional situation; a second section, of 16 items, aimed at assessing professionals on the impact that the current health crisis and state of alarm have generated on the usual development of social services; a third section related to the knowledge of the protection measures that have been carried out in the different entities (8 items); a fourth section aimed at analysing the professional and personal situation with the current situation (13 items), which is the object of analysis in this paper; and a fifth and final section related to the degree of adequacy of the measures aimed at the care of the vulnerable population (24 items) (See Appendix A).

### Description of the Sample

With regard to the profile of the social service professionals who have completed the questionnaire, it should be noted that the majority are women (83%) and that more than 70% of the sample is in the age group 35–59. Just over half of the sample are married (51.3%) and just over a third are single (38.6%), with children in slightly more than half of the cases.

In terms of training, 70.5% are university graduates, 73% of whom are social workers. In most cases (68%) they work in public institutions, compared to 24% of those working in private or subsidised institutions (7.7%).

With regard to the experience of social services professionals in the workplace, 41.5% of the sample have been working for less than 10 years and 40.9% between 11 and 22 years. Regarding the role in their work, a majority of technical staff is observed (66.6%), while 19% carry out management, coordination or team leader functions. The rest of the functions are distributed among programme managers, administrative functions and other functions.

## 3. Results

In order to facilitate the study of the results, factor analyses have been carried out, with varimax rotation, of sections two, four and five (measured on a likert scale) to reduce the number of items and work with this information in an aggregate way. The resulting factors have been transformed into a 0–100 scale to facilitate their presentation.

Table 1 shows the internal consistency of the dimensions that have been obtained through the factorial analysis measured through Crombach’s Alpha.

Table 2 shows the items that make up the dimension linked to the fourth section of the questionnaire, which analyses the perception of professionals in relation to the development of their work during the health crisis and the state of alarm, as well as the way in which the new work dynamic has affected them personally, with special emphasis on those that make up the factor known as “teleworking”.

In relation to the socio-demographic characteristics of professionals who telework in the field of social services, it is observed (Table 3) that men use this resource to a greater extent than women (64.37 compared to 54.99). On the other hand, in relation to age, there is a greater tendency to work telematically as age increases, fundamentally in the 35–59-year age group, and this decreases considerably in the last age group from 60 to 71. Regarding the marital status, separated and divorced people use this resource to a greater extent than married people and, finally, a greater use of this resource is seen as the level of training increases, except in the case of professionals with secondary studies, who do not follow this pattern. Although, it should be clarified that they represent very few cases in the sample.

However, in addition to the socio-demographic profile of the professionals who telework in social services, attention should be paid to the perceived effectiveness of this mode of work. In this sense, men perceive a better evaluation of teleworking, almost 10 points higher than that of women. With regard to age, the interval which best values this measure is from 47 to 59 years of age, which is also the group which makes the greatest use of this medium. On the other hand, a better evaluation is given among professionals with children, as well as among married and divorced people. In relation to the level of training, the professionals who most value this resource are those with postgraduate studies (masters and doctorate).

Table 4 analyses the profile of the labour situation. Given the nature of social work, there is a greater presence of teleworking (although without major differences) in other degrees, the same would be true of the perceived level of efficiency of telematic work, which is greater in other degrees. Telework is much more frequent among full-time workers, with a permanence in the workplace of between 11 and 33 years and belonging to public entities. However, in view of the position held, a greater implementation of this medium is observed among the functions linked to administrative tasks, as well as management, coordination and team responsibility.

On the other hand, regarding the perception of professionals about the efficiency of teleworking, those who value this resource most are full-time workers and those who have been working in the organisation for the longest time (between 23 and 44 years). In terms of the position they hold, the professionals who most value this resource are those who carry out management, coordination and team responsibility functions, followed by those who carry out functions as programme managers.

In Table 5, of the two items that make up the “telework” factor, the first one is used in a disaggregated way “I am teleworking at home and I am going to the workplace someday”, to see more specifically the socio-demographic and labour profile of the people who are teleworking in the social services. Similarly, in order to get a clearer idea of this worker profile, a variable is constructed that includes the two extreme options of the likert scale (recoded to 0–4), where 0 means not using this means at all and 4 means using it completely, thus omitting the intermediate options of the scale, which are the ones that represent the least cases, and thus grouping 70.9% of them.

As shown in Table 6, in relation to the assessment of the professional/personal situation in the face of the health crisis, it can be seen that those people who were teleworking experienced to a greater extent a feeling of being overcome by the situation, and this despite their perception of having had more resources and adequate measures to deal with the situation than those who continued to work in person.

On the other hand, those who teleworked had a poor perception (below 50 on the scale) of the degree of adequacy of the measures implemented for the care of the vulnerable population. Although, among those who continued their face-to-face activity, a lower degree of adequacy of these measures was even observed. Collaboration and socio-health follow-up measures were also perceived to be more effective among those who were teleworking.

Both groups of workers, both those who were teleworking and those who continued to work face-to-face perceived a high degree of worsening of the conditions of the vulnerable population with the state of alarm. Although, this worsening was perceived more strongly among those who had to telework.

With regard to the impact of COVID-19, people who switched to teleworking perceived that their functions had been more affected by the state of alarm than those who continued with face-to-face work, but this effect on functions was high in both cases. The people who were teleworking also perceived a greater degree of preparation and effectiveness of the measures implemented.

Likewise, with regard to the measure of teleworking, this is perceived as more effective by those people who use this route to carry out their work (50.06 compared to 31.26) although this score is still on the average of the scale, so it cannot be said, on the basis of the results obtained, that the participants themselves considered the teleworking mode to be particularly effective in the context of social services. Although, it cannot be observed either that this was valued as ineffective by those who were able to adapt their activity to it.

## 4. Discussion

The health crisis caused by COVID-19, with the effects suffered by the confinement and to prevent the spread of the virus, has given rise to a situation in which the work carried out by social service professionals who have gone from doing face-to-face work to online and telephone teleworking with the vulnerable groups who have suffered most from this pandemic is valued positively already signaled [29]. This situation has affected the occupational health of professionals from different circumstances. It has not been easy, as mentioned, since it has been necessary to adapt on-site intervention to remote intervention they have already pointed out [30].

Telework is a form of work which is increasingly recognised and expanding and would probably be implemented in the not too distant future. This working practice has become evident during the health crisis in some companies and public institutions and in the social services as previously mentioned. This situation has affected the occupational health of professionals from different circumstances. With the pandemic, teleworking is a labour reality that will have to be legally regulated through workers’ collective agreements they have already specified [31].

In the current globalized information society, it means changes in our daily lives. Telework is one of the possibilities offered by ICTs (Information and Communication Technologies) already pointed out [3]. ICTs have been “a preponderant factor during this process, facilitating communication throughout the planet; with the deployment of the internet, tools for collaboration in real time were designed and implemented, synchronising the movements of a dynamic world” they have already highlighted [32].

With regard to the use of technologies, special attention needs to be paid to the age of people who are teleworking, having found that the social services staff who are best suited to teleworking are those between 35 and 59 years of age. In contrast, teleworking is significantly reduced for people aged 60–71. Age, a variable linked in most cases to work experience in the same entity, seems to condition to a large extent the preference for teleworking. This data seems logical given that age constitutes a risk factor in the prognosis of curing the disease, this population group being considered more vulnerable to the coronavirus. The older the person, the greater the probability of suffering from chronic diseases and pathologies that can negatively affect the development of the disease. However, the lesser existence of this form of teleworking among professionals aged between 60 and 71 could be due to the lesser familiarity with this type of telematic resources (digital divide).

The situation caused by the health crisis has meant that urgent action has had to be taken and, in some cases, improvised, as the work situation has had to be adapted to an online format in a few days.

Once the initial state of alarm and home confinement of the population has been overcome, it seems that teleworking is still an option in some sectors for work carried out in person. For teleworking to be applied efficiently, everything must be highly organised and planned, and it requires greater effort and learning on the part of the workers. “The risks of contagion and the greater ease of work-family conciliation at a time when family demands are increased by the presence of all family members in the home. Moreover, this activity offers the opportunity to practice and learn about these new ways of working” [33].

Therefore, some telework experiences related to Social Services indicate the impossibility for most of these professionals to telework full time due to the nature of their duties. Nevertheless, it offers an opportunity to make the work that is al-ready being developed in a face-to-face way compatible with other forms of telematic work, which could mean an increase in the efficiency of the use of computer programs to support the intervention (to access case files, take notes, make evaluations, take photos, manage appointments etc.), which was observed to increase the precision of the work they have already pointed out [34].

Teleworking undoubtedly represents one of those opportunities that this crisis is highlighting, showing promising possibilities in the field of family reconciliation, productivity and for the environment, although the social contact with users that is so important for social service professionals is being lost. With teleworking, it will be more complicated to maintain the interpersonal relationships which existed before the current pandemic, although as months go by, these relationships will gradually recover.

Authors, such as [35,36] state that employees who telework tend to experience higher levels of satisfaction than those who do not. However, given the nature of social work (majority degree in the sample) and social services, the results of this work show a greater degree of overloading of professionals who telework, since they experienced to a greater extent than workers who were present the feeling of being overwhelmed by the situation, despite their perception of having had more resources and adequate measures to deal with it than those who continued to develop their activity in a face-to-face manner. In this line, teleworking could be seen in itself as one more resource which could have increased the sense of protection of workers by reducing their exposure to risk situations (such as going to the workplace, using transport, etc.). Although, on the other hand, it is easy to discern that adapting to the transition of activity to telematic means has been able to generate a greater situation of overload, especially among workers who are not accustomed to teleworking (as discussed in the introduction to this article, the percentage of workers who used this route before the pandemic).

We can say that telework is a rapidly growing work practice, but its effects on the psychological well-being of employees have been little studied. A particular problem for remote workers is the potential loss of social support, and while this can be provided electronically, how it affects worker welfare and performance is a concern that needs to be addressed both by research and by organizations where there is a significant amount of teleworking. However, this is a highly complex subject of study, as both the positive and negative effects of communication and interpersonal relations through telematic means must be taken into account, as well as the variables of individual differences such as personality and gender which underlie the motivations and experiences of telework. In addition, social support from home can be as important as support from the workplace, and how these combine is difficult to predict they have already pointed out [37].

Other authors, such as [38] point out that social service professionals have had to design, plan, implement and follow up on measures taken during the state of alert that included social dialogue and collective bargaining. Organizations can mitigate the detrimental effects associated with telework by introducing greater clarity in job description, structures and communication, and by structuring organizational practices to improve communication and feedback.

In this line, both groups of workers, those who were teleworking and those who continued with the face-to-face work, perceived in a high degree a worsening of the conditions of the vulnerable population with the state of alarm. Although this worsening was perceived with greater force among the people who had to telework, which could be due to the fact that they were not able to attend to their users as they had been doing, face-to-face. This situation may have led them to feel more strongly that the conditions of these people would have been worse than how it was perceived by the people who were able to continue with direct care.

While, a considerable number of employees worldwide are forced to work from home due to the COVID-19 crisis, it is important to highlight how the current way of doing professional work through teleworking is being managed. Therefore, it examines employees’ perceptions of teleworking in various aspects of life and career, distinguishing between typical and widespread teleworking during the COVID-19 crisis they have already noted [39].

The perceived effects of telework on other facets of respondents’ personal and professional lives are largely in line with the findings of previous studies. For example, many positive characteristics (e.g., increased efficiency and reduced risk of burnout) have been attributed to teleworking, while at the same time the possible negative effects on promotion opportunities and working relationships have been highlighted.

In this sense, the results of this study highlight that the perception of the poor development of some protective measures in the workplace at the initial moment of the state of alarm may have influenced the option of working telematically they have already pointed out [1], at least in cases where this option has been taken freely by the workers and not imposed by the organization. This fact may have conditioned the choice of telematic work as opposed to face-to-face work, since, given the lack of perception of safety in the workplace, professionals may have chosen to work from home.

Finally, the author [40] indicates that telework is positively related to commitment and negatively to intentions to renew personnel, so that a greater degree of telework is associated with a greater commitment to the organisation and weaker intentions to renew personnel. The positive effect of teleworking revolves around reduced work pressure and role conflict, as well as increased autonomy. The negative effect of teleworking is expressed through increased role ambiguity and reduced support and feedback.

## 5. Conclusions

It should be clarified that the results and conclusions of this work are close to a social services approach to telework. While, it has sought to achieve maximum representation of the Spanish territory, the conditions in which the field work has been carried out do not 100% guarantee the generalization of the results to the entire population of professions in this field, so the conclusions presented relate to the sample as a whole.

In relation to the majority profile of professionals working telematically in the field of social services and that they conform the sample of this work, the results show that this modality would be more prevalent among older people, married and divorced, with children and working in public institutions.

It should be pointed out that social services staff who were teleworking reported to a greater extent a feeling of being overwhelmed by the situation, despite considering that they had had more resources and adequate measures to deal with the situation than those who continued to work face-to-face.

For their part, social service personnel expressed a feeling of inadequacy in the measures implemented by governments to assist vulnerable populations. With regard to those who continued to work in person, they felt that the measures implemented had not been adequate.

Both the people who were working telematically and those who were working physically noted a worsening of the conditions of the vulnerable population during the state of alarm, although this was perceived to a greater extent by the personnel who were teleworking.

Finally, with regard to the effect of COVID-19, people who adapted to teleworking perceived that their tasks had been more affected by the state of alarm than those who continued to carry out their work in a face-to-face manner. Although the fact of teleworking as a measure for adapting to this context has been perceived more effectively by the personnel themselves who were teleworking, that is, once people make use of teleworking their assessment is more positive than the perception, they might have had without having made use of this medium.

In this way, the results have shown how the COVID-19 pandemic has greatly affected the way of working in social services, which was practically all face-to-face work. As an alternative, teleworking has allowed for an adaptation to this crisis and a greater flexibility in terms of work-life balance, and has shown how many of the activities carried out by social service professionals have been able to be developed in a non-presential manner, which could be a powerful way of speeding up the system. However, other consequences should also be taken into account, such as longer working hours, the feeling of “not being able to disconnect” from work, the possible psychosocial risks derived from this situation of teleworking, the impact of legislative reforms, the possible digital divide with older workers, consideration of gender differences, among others. It would be necessary to continue exploring the possible benefits and disadvantages of increasing teleworking in a widespread way in social services in a post-pandemic context.

## Figures and Tables

**Table 1 ijerph-18-00725-t001:** Factorial analysis.

Questionnaire Dimensions	Factors Obtained	No. of Items That Compose It	Crombach’s Alpha (Internal Consistency)
Assessment of the impact of the COVID-19 on professional development.	1. Degree of preparedness and effectiveness of measures implemented.	8	0.822
	2. Degree to which functions are affected by the state of alarm.	4	0.795
	3. Efficiency Teleworking ^1^	3	0.722
Assessment of the impact on the professional/personal situation.	1. Feeling overwhelmed by the situation.	5	0.777
	2. Appropriate resources and measures to address the situation.	6	0.740
	3. Teleworking	2	0.726
Assessment of the adequacy of measures to assist the vulnerable population.	1. Worsening conditions of vulnerable populations in a state of alarm.	9	0.947
	2. Degree of adequacy of measures to assist vulnerable populations.	8	0.862
	3. Effectiveness of collaboration and socio-health monitoring measures.	4	0.774

^1^ The efficiency of telework is evaluated through the efficiency perceived by professionals through the three items that make up this factor (In my job, telework has allowed to develop my professional work normally; I have had sufficient means to telework during the development of the crisis; I have had the necessary training and instructions to be able to develop my work in a telematic or non-face-to-face way).

**Table 2 ijerph-18-00725-t002:** Factorial analysis of the professional/personal situation assessment.

Factors	Assessment of Professional/Personal Situation (Items Making Up the Synthetic Index)	Crombach Alpha
1. Emotional difficulties and feeling overwhelmed by the situation.	I have often felt like crying these days.	0.777
	Throughout these days, discussions with social service colleagues have increased.	
	During these weeks, I have worked many more hours beyond my working hours.	
	It can be said that there have been times when I have felt overwhelmed by the situation.	
	I have often felt powerless these days. I have often felt supported and understood by the users of social services in the face of the difficulty of the situation.	
2. Appropriate resources and measures to address the situation.	I have had the necessary protective equipment to do my job.	0.740
	My professional mobility has not been affected. I have been able to travel to my workplace without any problems.	
	I have had the basic training to face my work during this time.	
	At my workplace I have been properly accredited to do my job during the alarm period.	
	In general, I have found support from my colleagues in solving the problems I have faced these days.	
3. Teleworking.	I’m teleworking at home and going to work one day.	0.726
	Despite teleworking, I know that at any time I can be called back to work.	

**Table 3 ijerph-18-00725-t003:** Socio-demographic profile of telework in social services and perception of the effectiveness of this resource.

		N	Teleworking % (Eactor)	Perception of Teleworking Efficiency
Sex	Man	93	64.37	47.99
	Woman	466	54.99	40.91
Age	From 23 to 34 years old	125	47.58	41.74
	From 35 to 46 years old	228	57.01	37.71
	From 47 to 59 years old	193	63.30	57.01
	From 60 to 71 years old	14	46.88	46.67
Children	Yes	303	61.12	43.26
	No	257	51.14	40.44
Civil status	Single	216	50.92	39.65
	Widower	6	62.50	59.72
	Divorced/Separate	51	62.12	48.52
	Married/domestic partnership	287	59.99	42.14
Level of studies	Secondary	12	58.93	40.28
	University students	395	54.15	39.54
	Master’s degree	133	62.01	46.71
	Doctorate	20	67.86	62.04

**Table 4 ijerph-18-00725-t004:** Profile of the employment situation.

		N	% of Teleworking	Perception of Teleworking Efficiency
Social Work and Other Degrees	Social Work	410	55.83	39.72
	Other Degrees	150	58.66	48.36
Employment situation	Volunteering	10	62.50	34.26
	Part-time work	59	48.26	36.63
	Full-time job	491	57.47	42.74
Time in years of work	0–10 years	210	53.01	43.09
	11–22 years	207	59.62	38.21
	23–33 years	73	62.72	50.98
	34–44 years	16	48.21	45.24
Type of organization	Private	134	47.65	44.44
	Concerted	43	42.92	31.94
	Public	381	61.48	42.34
The position you hold	Management functions, coordination, team responsibility.	107	61.68	51.84
	Program Manager	47	41.79	48.72
	Technical	373	56.69	39.40
	Administrative tasks	8	71.88	31.25
	Other functions	25	60	30.95

**Table 5 ijerph-18-00725-t005:** People who do not use telework at all and people who telework completely (values 0 and 4).

“I’m Teleworking at Home and Going to Work One Day” (P4_12_Professional_Situation_Recco_Personnel_in 0–4.)					
		N	%	% Valid	% Accumulated
Valid	0	174	31.1	36.5	36.5
	1	38	6.8	8.0	44.4
	2	46	8.2	9.6	54.1
	3	55	9.8	11.5	65.6
	4	164	29.3	34.4	100.0
	Total	477	85.2	100.0	
Lost	System	83	14.8		
Total		560	100.0		

**Table 6 ijerph-18-00725-t006:** Workers’ perception according to whether they work in person or telematically.

Dimensiones	Scale Factors 0–100	Teleworking	Face-to-Face Work
Assessment of the professional/personal situation in the face of the health crisis (P4)	Feeling overwhelmed by the scale of the situation.	58.54	50.71
	Adequate resources and measures to address the situation.	68.50	55.91
	Teleworking.	94.59	16.96
Assessment of actions aimed at vulnerable populations (P5)	Worsening conditions of vulnerable populations in a state of alarm.	77.37	66.84
	Degree of adequacy of measures to assist vulnerable populations.	43.95	38.04
	Effectiveness of collaboration and socio-health monitoring measures.	65.01	54.61
COVID-19 impact assessment (P2)	Degree of preparedness and effectiveness of measures implemented.	49.51	42.83
	Degree to which functions are affected by the state of alarm.	84.12	77.05
	Teleworking efficiency.	50.06	31.26

## Data Availability

Not applicable.

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
