# Peer review of "Telework and Social Services in Spain during the COVID-19 Pandemic"

_ijerph, 2021, doi:10.3390/ijerph18020725_

Round 1

Reviewer 1 Report

The paper addresses an actual and relevant topic. The study is supported by a survey´s data that describes how social service professionals had to deal with the pandemic and their experience with teleworking. From my perspective, this is the main value of this study, to describe the current situation of social service professionals in the middle of the pandemic, specifically regarding the impact on the usual development of social services, knowledge about protection, professional and personal situation, and adequacy of the care of the vulnerable population. This knowledge as a description is important by itself. I don’t understand why the data analysis tries to go beyond that. In this line, why the author conducts a correlational and a factorial analysis? Why reduce the number of items? (Items were eliminated by low Cronbach’s alpha?).

Related to the latter, there is not literature developed that support this study.  Variables are not properly defined and because of that, the framework involves different constructs not included in this study.   For example, it is mentioned variables such as psychological well-being, social support, worker welfare and performance, satisfaction, commitment, autonomy, etc.  In other words, the literature review is not properly linked with the methodology and results.

On the other hand, in the “Materials and Methods” section, the information is not complete.  There is not a procedure section, neither ethical considerations taking into account. The questionnaires were applied on-line? Or it was applied collectively in the institution?. The questionnaire or survey was build for this study?. It may include it in the appendix.

Due to the known differences in the public and private sectors, why include both sectors in the sample? even more, considering that most of the participants belong to the public sector?  (Only 7,7% belongs to the private sector) . This variable may be controlled. If the study is not only descriptive, the occupation, kind of job, position, and age should be controlled. How many institutions were including? From what cities? All this information needs to be completed.

The objective is written in different ways in the paper. I suggest unifying.

In the line 177 the author says “…the aim of this research has been to analyze how telework has been carried out in the field of social services and what characteristics define the professionals who have carried out their work through this modality in Spain during the first wave of the Covid-19”. I think that the first part of this objective is adjusted to the presented paper, however, in the second part, more than characteristics that define these professionals in Spain; they are the characteristics of the studied sample.

The objective written in the abstract says “…to analyze the profile of the professional who teleworked in social services and, on the other hand, to analyze the perception of teleworkers of working conditions during this period, as well as the degree to which they have been affected by them depending on whether they work face-to-face or telematically”. The statistical analysis does not provide information on the accomplishment of this objective. Why make correlations between telework and socio-demographic and labor variables?.

Likewise, I suggest being careful with the conclusion extracted from the correlation analysis. For example: “Thus, it could be concluded that the older the person, the greater the probability of carrying out the work telematically. A similar situation would arise in the case of having children, the fact of having dependent minors increases the probability of carrying out the work in a non-presential manner, thus improving the possibility of combining the professional and family spheres”.

In the conclusions, more than “majority profile of professionals working telematically in the field of social services, are more prevalent among older people….” They are sample characteristics.

Likewise, the author asserted that the Covid-19 pandemic has greatly affected the efficiency of the way of working in social services. Efficiency was assessed? How?

The collected data for this study is relevant and allows to know important issues regarding this relevant topic. However, I suggest redirecting the study from the beginning in order to give it a more clear structure. According to my perspective, this research is descriptive and its results should be oriented to this approach. It is not useful to search correlations among variables that conceptually do not add value. I suggest re-thinking the statistical analysis.

Finally, although I´m not qualified to judge the English language and style, I recommend not using colloquial terms such as “open the door…” “….have not been immune to this impact either…” “…the door is open to …”  “…If we look a …“

The quotes should be adjusted. For example “Along these lines, [13] already pointed out that …” “Authors such as [28,29] state …”

I suggest revising some terms: The majority is married (51.3%)….  (It is almost the same proportion).

Author Response

Open Review 1

English language and style

( ) Extensive editing of English language and style required
( ) Moderate English changes required
( ) English language and style are fine/minor spell check required
(x) I don't feel qualified to judge about the English language and style

Yes

Can be improved

Must be improved

Not applicable

Does the introduction provide sufficient background and include all relevant references?

( )

( )

(x)

( )

Is the research design appropriate?

( )

( )

(x)

( )

Are the methods adequately described?

( )

( )

(x)

( )

Are the results clearly presented?

( )

( )

(x)

( )

Are the conclusions supported by the results?

( )

( )

(x)

( )

Comments and Suggestions for Authors

The paper addresses an actual and relevant topic. The study is supported by a survey´s data that describes how social service professionals had to deal with the pandemic and their experience with teleworking. From my perspective, this is the main value of this study, to describe the current situation of social service professionals in the middle of the pandemic, specifically regarding the impact on the usual development of social services, knowledge about protection, professional and personal situation, and adequacy of the care of the vulnerable population. This knowledge as a description is important by itself. I don’t understand why the data analysis tries to go beyond that. In this line, why the author conducts a correlational and a factorial analysis? Why reduce the number of items? (Items were eliminated by low Cronbach’s alpha?).

  • As authors of the manuscript we have used factors because of the scale elaborated and removed the items because the correlations are low, but we introduce as Appendix B the values of the items.

Related to the latter, there is not literature developed that support this study.  Variables are not properly defined and because of that, the framework involves different constructs not included in this study.   For example, it is mentioned variables such as psychological well-being, social support, worker welfare and performance, satisfaction, commitment, autonomy, etc.  In other words, the literature review is not properly linked with the methodology and results.

  • Quotations have been included and the theoretical framework improved as you requested.

On the other hand, in the “Materials and Methods” section, the information is not complete.  There is not a procedure section, neither ethical considerations taking into account. The questionnaires were applied on-line? Or it was applied collectively in the institution?. The questionnaire or survey was build for this study?. It may include it in the appendix.

  • Appointments and information requested by you have been included. In addition, Appendix A has been included with the questionnaire carried out for the research.

Due to the known differences in the public and private sectors, why include both sectors in the sample? even more, considering that most of the participants belong to the public sector?  (Only 7,7% belongs to the private sector) . This variable may be controlled. If the study is not only descriptive, the occupation, kind of job, position, and age should be controlled. How many institutions were including? From what cities? All this information needs to be completed.

  • The private sector has been eliminated.

The objective is written in different ways in the paper. I suggest unifying. In the line 177 the author says “…the aim of this research has been to analyze how telework has been carried out in the field of social services and what characteristics define the professionals who have carried out their work through this modality in Spain during the first wave of the Covid-19”. I think that the first part of this objective is adjusted to the presented paper, however, in the second part, more than characteristics that define these professionals in Spain; they are the characteristics of the studied sample.

  • The objective has been unified in the manuscript as you requested and has been modified to fit the characteristics of the sample studied.

The objective written in the abstract says “…to analyze the profile of the professional who teleworked in social services and, on the other hand, to analyze the perception of teleworkers of working conditions during this period, as well as the degree to which they have been affected by them depending on whether they work face-to-face or telematically”. The statistical analysis does not provide information on the accomplishment of this objective. Why make correlations between telework and socio-demographic and labor variables?

  • The considerations made by you have been taken into account.

Likewise, I suggest being careful with the conclusion extracted from the correlation analysis. For example: “Thus, it could be concluded that the older the person, the greater the probability of carrying out the work telematically. A similar situation would arise in the case of having children, the fact of having dependent minors increases the probability of carrying out the work in a non-presential manner, thus improving the possibility of combining the professional and family spheres”.

  • The considerations made by you have been taken into account.

In the conclusions, more than “majority profile of professionals working telematically in the field of social services, are more prevalent among older people….” They are sample characteristics.

  • The considerations made by you have been taken into account.

Likewise, the author asserted that the Covid-19 pandemic has greatly affected the efficiency of the way of working in social services. Efficiency was assessed? How?

  • The information requested has been included as a note in table 1.

The collected data for this study is relevant and allows to know important issues regarding this relevant topic. However, I suggest redirecting the study from the beginning in order to give it a more clear structure. According to my perspective, this research is descriptive and its results should be oriented to this approach. It is not useful to search correlations among variables that conceptually do not add value. I suggest re-thinking the statistical analysis.

  • The considerations made by you have been taken into account.

Finally, although I´m not qualified to judge the English language and style, I recommend not using colloquial terms such as “open the door…” “….have not been immune to this impact either…” “…the door is open to …”  “…If we look a …“

The quotes should be adjusted. For example “Along these lines, [13] already pointed out that …” “Authors such as [28,29] state …”

I suggest revising some terms: The majority is married (51.3%)….  (It is almost the same proportion).

  • The considerations made by you have been changed.

Reviewer 2 Report

I want to congratulate the authors for such a brilliant contribution to the scientific knowledge of Social Services and its management in times of pandemic.
Telecommuting and Digital Social Work, which existed in many practices, has come to stay and promote closeness, empathy and active listening with the support of other tools.
I encourage the authors to continue investigating this topic and to delve into in a textbook to include in the training of Social Work and Social Services students.

Author Response

Thank you very much for your words and the academic recognition given to our research. As authors of the manuscript we will continue to research and take into consideration the proposal you put forward.

Reviewer 3 Report

The work presented is very interesting. It is worth exploring how the outbreak of the COVID pandemic and the measures that have been taken have affected social services and by extension the people who are the object of its intervention.

Congratulations to the authors for the excellent writing of the paper presented, as well as for the originality of the research proposal.

On the other hand, it is to value the important sample that they have achieved. Something that is extremely complicated and even more so among a sector such as social services professionals.

The presentation of the results is correct and easy to read.

However, there are some aspects to consider to improve the work presented:

- In the second paragraph of the introduction the authors incorporate numerous statements that need to be supported in the literature. As well as eliminating the suspension points in line 38.
- The statements that the authors pour out on page 3, line 115 (and following) correspond more to the discussion or conclusion than to presenting the state of the art. It is therefore worthwhile placing them elsewhere.
- The statement of opinions in the introduction should be avoided and their inclusion in the discussion should be considered.
- Likewise, it is relevant to support with bibliographic references the statements made at different moments of the introduction.
- It would be interesting to reflect the regions from which the participants in the study come. The development of social services is not similar in all Spanish regions.
- It is not shown how the ethical principles have been fulfilled in the research or how the confidentiality and anonymity of the participants has been assured.

Round 2

Reviewer 1 Report

The integration between the conceptual and empirical sections was improved. This was one of the most important issues that should be corrected.
The document needs proofreading. For example, the words “pointed out” are written several times (see lines 136 to 173 and lines 337 to 357 only to give an example). Revise also minor details such as: “In the following table (Table 6)….”, it is enough to say “in table 6….”
When the author mentions “profile of professionals working telematically in the field of social services…” I still consider that this is the sample description more than a profile. It is not enough to say that all autonomous communities are represented to talk about a profile.
I think that the results of this study are not generalizable because the sample (neither the sample type) is not representative of the whole country. If it is representative, please describe the sample size of each autonomous community, demonstrating its representativeness.
If the sample is not representative is not correct to say in the conclusion: “ …. this modality would be more prevalent among older people, married and divorced, with children and working in public institutions” as a profile. I suggest to rethink this aspect as the objective of this paper and leave this information as a sample description.
On the other hand, I suggest adding the cities name where the sample was taken.
According to my point of view, the new references include in Materials and Methods are not needed. I think that more that contribute affect this section.

In the response letter, the authors say that the private sector was eliminated. Then, why the sample size is the same? Only the label was eliminated?

Author Response

Open Review

(x) I would not like to sign my review report

( ) I would like to sign my review report

English language and style

( ) Extensive editing of English language and style required

(x) Moderate English changes required

( ) English language and style are fine/minor spell check required

( ) I don't feel qualified to judge about the English language and style

                Yes         Can be improved             Must be improved          Not applicable

Does the introduction provide sufficient background and include all relevant references?

                (x)          ( )            ( )           ( )

Is the research design appropriate?

                ( )           ( )            ( )           (x)

Are the methods adequately described?

                (x)          ( )            ( )           ( )

Are the results clearly presented?

                ( )           (x)           ( )           ( )

Are the conclusions supported by the results?

                ( )           (x)           ( )           ( )

Comments and Suggestions for Authors

The integration between the conceptual and empirical sections was improved. This was one of the most important issues that should be corrected.

The document needs proofreading. For example, the words “pointed out” are written several times (see lines 136 to 173 and lines 337 to 357 only to give an example). Revise also minor details such as: “In the following table (Table 6)….”, it is enough to say “in table 6….”.

  • All the issues identified have been reviewed. In particular, on page 6, in the first line of the second paragraph, The following table (Table 2) is replaced by The table 2 shows. Also on page 9, in the first line of the first paragraph, In the following table (Table 6) is replaced by As shown in table 6. Finally, the words "pointed out" have been changed throughout the manuscript.

When the author mentions “profile of professionals working telematically in the field of social services…” I still consider that this is the sample description more than a profile. It is not enough to say that all autonomous communities are represented to talk about a profile. I think that the results of this study are not generalizable because the sample (neither the sample type) is not representative of the whole country. If it is representative, please describe the sample size of each autonomous community, demonstrating its representativeness. If the sample is not representative is not correct to say in the conclusion: “ …. this modality would be more prevalent among older people, married and divorced, with children and working in public institutions” as a profile. I suggest to rethink this aspect as the objective of this paper and leave this information as a sample description.

  • The full first paragraph regarding the sample has been introduced on page 12 in the conclusions section. The results of the study cannot be generalised as indicated and nothing else has been added with respect to the profile because we believe that it is already quite clear and it seems to us to be very repetitive to put every time a profile of the sample in the manuscript, because this is already clarified.

On the other hand, I suggest adding the cities name where the sample was taken.

  • The analysis has been coded by Autonomous Communities.

According to my point of view, the new references include in Materials and Methods are not needed. I think that more that contribute affect this section.

  • All these references have been removed from the manuscript.

In the response letter, the authors say that the private sector was eliminated. Then, why the sample size is the same? Only the label was eliminated?

  • The cases have not been definitively eliminated and the disaggregated analysis is not carried out because there is very little representativeness as you indicated in the first review.